# Construction of a Versatile, Programmable RNA-Binding Protein Using Designer PPR Proteins and Its Application for Splicing Control in Mammalian Cells

**DOI:** 10.3390/cells11223529

**Published:** 2022-11-08

**Authors:** Yusuke Yagi, Takamasa Teramoto, Shuji Kaieda, Takayoshi Imai, Tadamasa Sasaki, Maiko Yagi, Nana Maekawa, Takahiro Nakamura

**Affiliations:** 1EditForce, Inc., Fukuoka 819-0395, Japan; 2Faculty of Agriculture, Kyushu University, Fukuoka 812-8581, Japan

**Keywords:** RNA, pentatricopeptide repeat protein, splicing

## Abstract

RNAs play many essential roles in gene expression and are involved in various human diseases. Although genome editing technologies have been established, the engineering of sequence-specific RNA-binding proteins that manipulate particular cellular RNA molecules is immature, in contrast to nucleotide-based RNA manipulation technology, such as siRNA- and RNA-targeting CRISPR/Cas. Here, we demonstrate a versatile RNA manipulation technology using pentatricopeptide-repeat (PPR)-motif-containing proteins. First, we developed a rapid construction and evaluation method for PPR-based designer sequence-specific RNA-binding proteins. This system has enabled the steady construction of dozens of functional designer PPR proteins targeting long 18 nt RNA, which targets a single specific RNA in the mammalian transcriptome. Furthermore, the cellular functionality of the designer PPR proteins was first demonstrated by the control of alternative splicing of either a reporter gene or an endogenous *CHK1* mRNA. Our results present a versatile protein-based RNA manipulation technology using PPR proteins that facilitates the understanding of unknown RNA functions and the creation of gene circuits and has potential for use in future therapeutics.

## 1. Introduction

The emergence of genome/transcriptome data in the last two decades has greatly facilitated our understanding of biological organisms and their uses. The recent establishment and popularization of genome editing technologies will provide new opportunities for rational genome engineering in various fields [1,2,3]. Despite the advances in DNA manipulation technologies, the development of RNA manipulation strategies lags further behind, although the importance of RNA and its function has been realized by the discovery of massive complexed alternative splicing and a large number of non-coding RNAs [4,5]. In addition, the recent concerns regarding off-target genome editing technology, which permanently alters genomic DNA, have increased the demand for elaborate RNA manipulation technique(s) to enable conditional, flexible, and reversible changes in the expression of genetic information.

RNA manipulation can be divided into two classes: nucleotide-based and protein-based. Nucleotide-based technology includes antisense oligonucleotides (ASOs) and RNA interference (RNAi), which have been widely used for knock-down or inhibition of target RNA functions in basic biological research and various therapeutics [6,7]. However, this serves as a guide for the targeted position, and the subsequent flow relies on endogenous factor(s) or cellular mechanisms [8,9]. Conversely, recently established RNA-guided, RNA-targeting CRISPR/Cas-based systems can be used as conjugates with additional functional domains, as shown by knockdown, visualization, and splicing control, as well as RNA epigenetic modifications in the cell [10,11,12,13]. This new CRISPR/Cas-based RNA manipulation can produce more efficient and effective RNA manipulation than ASO or siRNA, whereas the nucleotide-based technology depends on base pairing, which is prone to off-targeting, as shown in RNAi- or DNA-targeting CRISPR/Cas systems.

However, protein-based technologies present promising advantages, as seen in various cases of cellular RNA processing that have been managed by RNA-binding protein factors. The manipulation of a target RNA can be achieved in a desired manner by fusing a sequence-specific RNA-binding protein with an enzymatic effector domain [14]. A well-known example is the use of the MS2 protein, which specifically binds RNA with a particular stem–loop structure and particular sequences [15]. Various applications have been developed, such as RNA degradation by fusing SMG6 with the MS2 protein [16]. However, the MS2 target sequence should be inserted into the RNA of interest because the sequence specificity of MS2 cannot be changed, indicating that this technique is difficult to apply to endogenous RNA sequences. This disadvantage was partially resolved by Pumilio/fem-3 mRNA-binding factor (PUF). The PUF protein has a modular structure, which repeats eight to nine PUF motifs, and its sequence specificity is determined by the combination of two particular positions of amino acids within PUF motifs [17,18]. Various RNA manipulation tools have been developed using PUF proteins, including RNA visualization, degradation, and translational enhancement [19,20,21]. However, the use of PUF protein has been restricted, because the PUF protein recognizes only eight bases, and binding sequence alteration is not fully programmable.

These disadvantages in protein-based technologies may be resolved by using pentatricopeptide repeat (PPR) protein, which is a sequence-specific RNA-binding protein whose gene family is extraordinarily expanded in land plants [22]. The PPR protein gene is nuclear-encoded, but the protein is exclusively located in the mitochondria and chloroplasts and is involved in various aspects of RNA metabolism, including RNA processing, mRNA stability, and RNA editing, through sequence-specific RNA binding. The PPR protein consists of modular structures of repeating PPR motifs, as seen in PUF proteins, although the repeat number is highly diverse (2–27 repeats). The mechanism of sequence-specific recognition of RNA by the PPR protein has been elucidated, indicating that the target RNA length is determined by a number of PPR motifs, with one PPR motif corresponding to one nucleotide. The target RNA nucleotide is defined by the combination of amino acids at three particular positions (the 2nd, 5th, and 35th) within the motif [23,24]. The PPR-RNA recognition rule, known as the PPR code, suggests that the PPR protein is an attractive resource for engineering designer sequence-specific RNA-binding proteins.

Although several attempts have been made, many challenges remain in its use as a designer RNA-binding protein. The first is the amino acid context of the PPR motif. Although frequently observed amino acid combinations have been observed in the PPR code-generating residues (the 2nd, 5th, and 35th), the other amino acid sequences of natural PPR proteins in plants are highly degenerated. To date, various PPR scaffolds have been tested using consensus or semi-consensus sequences of natural PPR proteins [25,26,27,28] (Table 1). The synthetic PPR proteins were designed with 8 to 14 PPR motifs and have a range of target binding affinities between 7.5 nM and 370 nM of the *K*_D_ (Table 1), whereas the sequence specificity and/or stability of synthetic PPR proteins has not been addressed in living cells to date. Another challenge is determining the length of the target RNA. The natural PPR protein has 11 PPR motifs, that is, 11 nt recognition, on average, which is sufficient to target a single RNA in mitochondrial and chloroplast genomes of several hundred base pairs. However, 16 nt targeting is indispensable for its use in the mammalian transcriptome, based on the formula [(1/4)^16^ = 1/4.2 billion], against the 3 billion base pairs of the human genome. However, synthetic PPR proteins contain 14 motifs that have been tested thus far. Importantly, RNA control by designer PPR proteins has to be experimentally evaluated in living mammalian cells.

To address the possibility of developing designer PPR proteins as part of a new protein-based RNA manipulation technique, we evaluated the characteristics of dozens of designer PPR proteins. High-throughput construction and evaluation systems for designer PPR proteins have been developed for this purpose. With improvement of the PPR scaffold, we have demonstrated that designer PPR proteins with 18 nt RNA recognition can be prepared in a highly successful production ratio (91%; 21 out of 23 proteins)—proteins with higher RNA binding affinities and specificities than previous synthetic PPR proteins. Furthermore, the use of designer PPR proteins in splicing control of endogenous mRNA in human cultured cells has been demonstrated. Therefore, the use of designer PPR proteins as versatile protein-based RNA manipulation tools in basic and applied biology is proposed.

## 2. Materials and Methods

### 2.1. Construction of the Intermediate Plasmids Containing Two PPR Repeats for the Golden Gate Assembly of the Designer PPR Protein Gene

For the golden gate assembly of the designer PPR protein with 18 PPR motifs, an intermediate plasmid comprising two PPR motifs was chemically synthesized (Azenta Life Science, Chelmsford, MA, USA). The sequence information regarding the SCD, PPR2.0, and PPR3.0 scaffolds for the respective base recognitions (adenine, cytosine, guanine, and uracil) is shown in Table 1 and Appendix A. The plasmids for the respective scaffolds were composed of 144 plasmids divided into nine groups (pL-1 to pL-9). Each pL group contained 16 plasmids for all 16 di-nucleotide combinations recognized by the two PPR motifs. The capping motifs of N-CAP (MGNS) and C-CAP (ELTYNTLISGLGKAGRARDPPV) were attached to the N-and C-termini of the PPR repetition, respectively, to increase protein stability, as in cPPR [25]. N-CAP was fused to the PPR sequence inserted into the pL-1 vector (Appendix A). The first three C-CAP amino acids were fused with the PPR repeats in pL-9, and the remaining C-CAP sequence was introduced into the expression vector.

### 2.2. Construction of the Expression Vector

Prior to the golden-gate reaction for PPR motif assembly, extra domain sequences to be fused to the N- or C-termini of the PPR gene were integrated into the expression vector with the *Bpi*I site (Appendix A). The extra domain included a partial C-CAP sequence. Modified pET21 vectors (pETgg_a, pETgg_b, and pETgg_c) were used to express the PPR protein in *E. coli*. The expression vectors (pMgg_sp and pMgg_gfp) were used for the mammalian expression under the control of the CMV promoter.

### 2.3. Golden Gate Assembly of Designer PPR Gene

The designer PPR gene of 18 PPR motifs was assembled using the pL-1 to pL-9 plasmids and the expression plasmids via a golden-gate reaction with *Bpi*I and T4-DNA ligase (Appendix A). Twenty nanograms of each pL plasmid was mixed in 10 µL golden-gate reaction mixture, including 1 µL of 10× ligase buffer (New England Biolabs, Ipswich, MA, USA), 0.5 µL of *Bpi*I (Thermo Fisher Scientific, Waltham, MA, USA), 0.5 µL of Quick ligase (New England Biolabs), and 25 ng of the expression plasmid. The reaction was conducted in a thermal cycler, according to the following temperature cycles: 37 °C for 5 min and 16 °C for 7 min, for 15 cycles. After the reaction, 0.4 µL of *Bpi*I was added to the tube and further incubated at 37 °C for 30 min and 75 °C for 6 min. To degrade non-ligated DNA fragments, 0.4 µL of 1 mM ATP and 0.15 µL plasmid-safe nuclease (Epicentre-Lucigen, Middleton WI, USA) were mixed and incubated at 37 °C for 15 min. Then, 3.5 µL of the reactions were transformed into XL1-blue-competent cells (Nippongene, Tokyo, Japan). To confirm the size of the inserted PPR repeats in the expression plasmid, colony PCR was performed with Fw 5′-tacagggcgcgtcccattcgcca-3′ and Rv 5′-tcactataggggaattgtgagcgga-3′. The amplified fragments were analyzed using MultiNA (SHIMADZU, Kyoto, Japan). Sequence information for the PPR proteins used in this study and their RNA target sequences is shown in Appendix A. The amino acid sequences of all the fusion proteins are shown in the Appendix A.

### 2.4. Expression and Purification of Recombinant MS2 Protein

The expression plasmids encoding the His-tag fused MS2 protein gene were transformed into Rosetta 2(DE3) (Merck, Darmstadt, Germany). LB culture (100 mL) supplemented with 200 µg/mL ampicillin and 35 µg/mL chloramphenicol was inoculated with 1/100 volume of pre-cultured transformed cells and cultured at 37 °C until OD600 = 0.5. The recombinant proteins were induced with 0.1 mM IPTG at a final concentration of 15 °C for 16 h with shaking. After harvesting and centrifugation, the cell pellets were dissolved in ice-cold extraction buffer (20 mM Tris-HCl, pH 7.5, 500 mM NaCl, 2.5 mM MgCl_2_, 0.5% NP-40 (Sigma-Aldrich, St. Louis, MO, USA), 1 mM PMSF, 2 mg/mL lysozyme, and 2 µL 10 mg/mL DNase) and then sonicated. Cell debris was removed after centrifugation (twice at 15,000× *g*, 10 min, 4 °C). Half of the lysate solution was stored at −80 °C until use. For purification of His-tagged MS2 protein, 200 µL Ni-NTA slurry beads (QIAGEN, Venlo, The Netherlands) were added to the lysate, incubated with rotation at 4 °C for 1 h, and washed three times with wash buffer (20 mM Tris-HCl, pH 8.0, 500 mM NaCl, 0.5% NP-40, and 10 mM imidazole). The recombinant protein was eluted using elution buffer (20 mM Tris-HCl, pH 8.0, 500 mM NaCl, 0.5% NP-40, and 500 mM imidazole) and was dialyzed at 4 °C overnight in dialysis buffer (20 mM Tris-HCl, pH 8.0, 150 mM NaCl, 0.5% NP-40, 1 mM DTT, and 1 mM EDTA).

### 2.5. RNA-Binding Protein ELISA (RBP-ELISA)

The detailed protocol is described in the Appendix A. All sequences of the 5′-biotinylated synthetic RNA (30-mer; Greiner Japan, Tokyo, Japan) used in this study are shown in Appendix A.

### 2.6. Electrophoresis Mobility Shift Assay (EMSA)

For EMSA, the PPR protein was briefly purified with streptavidin beads followed by Ni-NTA beads. The resulting supernatants from 100 mL cultured bacterial cells were mixed with 100 µL streptavidin sepharose beads (Streptavidin Sepharose High Performance; Cytiva, Tokyo, Japan) and incubated at 4 °C with rotation for 1 h. The beads were washed three times with wash buffer containing 20 mM Tris-HCl (pH 8.0), 500 mM NaCl, and 0.5% NP-40. The protein was eluted using a biotin elution buffer (20 mM Tris-HCl, pH 8.0, 500 mM NaCl, and 2 mM biotin). Ni-NTA beads (200 µL) were added to the eluted protein solutions immediately and incubated at 4 °C with rotation. The buffer conditions for each step, including washing, elution, and dialysis, were the same as those mentioned above. The concentration of purified proteins was determined using the Pierce 660 nm Protein Assay Kit (Thermo Fisher Scientific, Waltham, MA, USA), according to the manufacturer’s protocols.

The biotinylated-RNA (0.2 pmol—the same synthetic RNA as that used in RBP-ELISA) and the purified protein at the indicated concentrations were mixed with 10 µL reaction buffer (20 mM Tris-HCl (pH 8.0), 150 mM NaCl, 0.5% NP-40, 1 mM DTT, and 1 mM EDTA). The reaction mixtures were resolved on a 6% polyacrylamide gel in 0.5× TBE buffer. The RNA and proteins in the gel were transferred to a nylon membrane (Hybond N+, GE Healthcare, Chicago, IL, USA) with 1× TG buffer using a semi-dry blotter. After blotting, the membrane was crosslinked and then blocked in membrane blocking buffer containing 6.4 mM NaH_2_PO_4_, 6.7 mM Na_2_HPO_4_, 125 mM NaCl, and 5% (*w*/*v*) sodium dodecyl sulfate. After blocking, the membrane was incubated in 10 mL blocking buffer including 0.5 µL streptavidin-HRP (ab7403) for 15 min with gentle shaking. The membrane was washed five times with 0.1× blocking buffer and equilibrated with buffer containing 100 mM Tris-HCl (pH 9.5), 100 mM NaCl, and 10 mM MgCl_2_. After removing the buffer, Immobilon Western Chemiluminescent HRP Substrate (Millipore) was added to the membranes. Chemiluminescent signals were detected using a ChemiDoc^TM^ Touch instrument (Bio-Rad, Hercules, CA, USA). The shifted and non-shifted band intensities were determined using Image Lab software (Bio-Rad, Hercules, CA, USA). The overall apparent *K*_D_ value was determined from the concentration of protein at which 50% of the RNA probe was bound as an indication of the RNA binding affinity.

### 2.7. Gel Filtration Assay

The SUMO-His-tag fused PPR2.0 and PPR3.0 proteins for the t24 target were expressed by a pET *E. coli* protein expression system. Soluble proteins were extracted using lysis buffer and purified using Ni-NTA, as described above. After purification, the SUMO-tag was removed by treatment with Ulp1, and dialysis was performed with ion-exchange buffer, including 50 mM Tris-HCl (pH 8.0) and 200 mM NaCl. Protein samples were added to the SP column and eluted by gradually increasing the NaCl concentration from 200 mM to 1 M. The PPR-containing fraction was separated with Superdex 200 increase 10/300 GL (Cytiva, Tokyo, Japan) with 25 mM HEPES (pH 7.5), 200 mM NaCl, and 0.5 mM tris(2-carboxyethyl)phosphine (TCEP). The amount of protein in each fraction was determined by measuring the absorbance at 280 nm.

### 2.8. Transfections

The splicing reporter assay plasmid for RG-6 was obtained from Addgene (#80167). HEK293T cells were cultured using Dulbecco’s modified Eagle’s medium (DMEM) (FUJIFILM Wako, Osaka, Japan), 10% fetal bovine serum (FBS) (Gibco^TM^, Thermo Fisher Scientific, Waltham, MA, USA), and penicillin–streptomycin (Gibco^TM^, Thermo Fisher Scientific, Waltham, MA, USA) at 37 °C with 5% CO_2_. For transfection, HEK293T cells were seeded into 24-well plates at 80,000 cells per well on the day before transfection and transfected with 500 ng RG-6 plasmid and 500 ng of the designer PPR-containing plasmid using FuGENE HD (Promega, Madison, WI, USA). The medium was changed after 24 h, and the cells were incubated for an additional 24 h at 37 °C with 5% CO_2_.

### 2.9. RNA Extraction, Reverse-Transcription, and RT-PCR

Total RNA was extracted using a Maxwell^®^ RSC simplyRNA Cell Kit (Promega, Madison, WI, USA), according to the manufacturer’s instructions. cDNA was generated in a 20 µL reaction mixture containing 5 µL total RNA (500 ng), 0.5 µL 100 µM oligo dT20 primer, 0.5 µL 10 mM dNTPs, 0.5 µL 0.1 M DTT, 0.5 µL 5× reaction buffer, 0.5 µL D.W., and 0.5 µL Superscript III (Invitrogen). The mixture was incubated at 50 °C for 30 min and then heated at 85 °C for 10 min to inactivate the reverse transcriptase. The cDNA was stored at −20 °C until use. To determine the influence of splicing efficiency by rPPR, PCR was performed with RG6Fw (5′-CAAAGTGGAGGACCCAGTACC–3′) and RG6Rv (5′-GCGCATGAACTCCTTGATGAC–3′) for the RG6 reporter assay, 5′-CHK1_F1 (CTCGGTGGAGTCATGGCAGTG–3′) and CHK1_R1 (5′-CATCTTGTTCAACAAACGCTCACG–3′) for the CHK1 exon 3 skipping assay, and MDM2_Ex3_Fw (5′-CGCGAAAACCCCGGATGGTGAG–3′) or MDM2_Ex6_Fw (5′-CATGATCTACAGGAACTTGGTAGTAGTC–3′) or MDM2_Ex7_Fw (5′-GAGAACAGGTGTCACCTTGAAGGTGG–3′) and MDM2_Ex12_Rv (5′-ATTCATTTCATTGCATGAAGTGCATTTCC–3′) for the MDM2 exon 9 skipping assay, using PrimeSTAR GXL DNA polymerase (Takara-bio, Shiga, Japan). The PCR product was analyzed using MultiNA (SHIMADZU, Kyoto, Japan) with a DNA 1000 kit (SHIMADZU, Kyoto, Japan). The ratio of exon skipping was calculated as the band intensity of the exon-skipped RNA divided by the sum of the exon-skipped and included RNAs.

### 2.10. Observation by Fluorescent Microscopy

Fluorescent images were captured using a Leica DMi-8 with FITC, RHOD, and DAPI filters for GFP, dsRED, and DAPI staining, respectively. For the RG-6 reporter assay, to determine the exposure time for imaging, only reporter-transfected cells were used to investigate the conditions under which the fluorescence intensities of FITC and Texas RED were comparable.

### 2.11. Construction of Stable Cells and Dox Induction

SV40 NLS-fused PPR^chk1a^ and PPR^sp6^ were cloned into a modified pSBtet-Pur vector (#60507; Addgene). The PPR expression plasmid and pCMV (CAT) T7-SB100 (#34879; Addgene) were transfected into HEK293T cells, which were selected using puromycin. Cells (1.5 × 10^5^) from each stable cell line were seeded into 6-well plates, and 0, 1, or 10 µg/mL doxycycline was added one day after seeding. After three days, total RNA was extracted using Maxwell (Promega, Madison, WI, USA). Exon skipping of CHK1 exon 3 was analyzed by RT-PCR. For analysis of cell viability, after three days of induction with 0 or 10 µg/mL Doxycycline (Dox), cells were collected and cell survival was determined by counting, using a Countess^®^ II FL (Thermo Fisher Scientific, Waltham, MA, USA) after trypan blue staining.

## 3. Results

### 3.1. The Construction Method for the Designer PPR Proteins That Bind to Target RNA Sequences

#### 3.1.1. Scaffolds of the Designer PPRs

Various concepts of designer PPR proteins were tested (Table 1). We began preliminary experiments regarding designer PPR proteins using the SCD scaffold because the SCD scaffold with 14 PPR motifs displayed the highest binding affinity [28]. The dPPR consists of the same amino acid context as the SCD scaffold [26,27]. However, the preparation of several designer PPR proteins occasionally failed at the DNA cloning and protein production steps, presumably owing to the repeated nature of the DNA sequence and protein aggregation, as seen in the production of recombinant proteins for endogenous natural PPR genes.

The RNA binding selectivity of the PPR motif is determined by amino acid species at the 2nd, 5th, and 35th amino acid positions, known as the PPR code (which positions are the same as the 1st, 4th, and ii positions described by Yagi et al. [23]). The VTN, VNS, VTD, and VND at the 2nd, 5th, and 35th amino acid positions were the most frequently observed PPR codes specifying adenine, cytosine, guanine, and uracil nucleotides, respectively. These representative PPR codes have also been utilized in various designer PPR proteins, including the SCD scaffolds. During large-scale PPR sequence analyses, the differences in amino acid contexts were observed at multiple positions in the PPR scaffold, when PPR motif sequences with the same PPR code were aligned (Appendix A). Considering the previous suggestion of the involvement of unknown positions of amino acids of the PPR scaffold in RNA binding capacity and to reduce the repetitiveness of PPR motif sequences with a protein at the nucleotide level, we decided to use the semi-consensus scaffolds for A, C, G, and U recognition by analyzing 322 canonical PPR motifs in Arabidopsis. In this study, the newly designed PPR scaffold was designated PPR2.0.

#### 3.1.2. Establishment of a High-Throughput Cloning System for Designer PPRs

In contrast to oligonucleotide-based technology, a disadvantage of protein-based technologies is their complicated construction method, which requires substantial expenditures of time and effort. The construction of proteins with modular structures, including TALE, PUF, and PPR genes, has been achieved using golden-gate cloning techniques [30,31,32].

The TALEN assembly containing 12 to 20 TALE repeats was conducted using a two-step reaction. First, a pre-step can be avoided by preparing an intermediate consisting of two or four TALE repeats (Figure 1a). The actual assembly of the protein containing 18 TALE repeats was conducted using an intermediate [30]. It has also been demonstrated that the designer PPR protein can be constructed using a similar two-step reaction using an intermediate containing three PPR motifs [32].

The target length of the designer PPR protein was determined with 18 nt recognition for use in mammalian cells, which is the same length as the designer TALE protein and theoretically sufficient to target a single RNA in the mammalian transcriptome. To construct a designer PPR protein containing 18 PPR motifs, we applied a similar golden gate assembly using two PPR motifs as an intermediate to reduce the total number of intermediates. DNA sequences of the 144 PPR motifs (=16 × 9) were chemically synthesized while maintaining degeneracy at the nucleotide level to avoid homologous recombination between DNA sequences encoding PPR motifs (Appendix A).

Using this method, four designer PPR protein genes with different target RNA selectivities were constructed using the PPR2.0 scaffold, in addition to the SCD scaffold as a control. The cloning efficiency of the four designed PPR genes of the PPR2.0 or SCD scaffolds was estimated by electrophoresis and sequencing by choosing three clones per PPR gene (Appendix A). The cloning efficiency of the PPR2.0 scaffold (11 out of 12) was better than that of the SCD scaffold (9 out of 12). The lower efficiency of the SCD scaffold may have been due to sequence homology among each PPR repeat (98%), resulting in intramolecular recombination during the cloning steps. This experiment also suggested that analysis of three clones is sufficient to obtain a correct PPR gene construct in the case of the PPR2.0 scaffold.

#### 3.1.3. Establishment of a High-Throughput Cloning System for Designer PPRs

Evaluation of RNA binding affinity and selectivity of the recombinant protein provides fundamental information. However, this also requires substantial time and effort, including purification, dialysis, and quantitation, and the use of low-throughput conventional analytical methods, such as electrophoretic mobility shift assay (EMSA) or surface plasmon resonance analysis using Biacore (Figure 1a).

To overcome this disadvantage, we designed a new method designated the RNA-binding protein ELISA (RBP-ELISA), which is a 96-well-plate-based ELISA-like assay that can be used to evaluate the RNA binding affinities and selectivities of expressed recombinant proteins using cell lysates without purification. To perform this method, 5′-biotinylated synthetic RNA containing the RNA sequence of interest was immobilized on a streptavidin-coated 96-well plate (Figure 1b). An RNA-binding protein fused with luciferase (luc) was added to the well to measure luminescence and determine the amount of protein bound to the immobilized RNA.

This method was evaluated using recombinant MS2 protein, which is a well-known RNA-binding protein. The MS2 protein was expressed in *E. coli* as a conjugate of Nano-luc and 6×His-tag. First, the recombinant protein was purified from half of the cell lysate and subjected to RBP-ELISA using the target RNA and non-target RNA (Appendix A). The luminescence signal was obtained only in the target RNA, depending on the dose of the recombinant MS2 protein (Appendix A). The remaining cell lysate was directly subjected to RBP-ELISA without purification. Dose-dependent specific binding signals were obtained only in the target sequence (Appendix A), as seen in the purified protein. This indicated that the RBP-ELISA had performed well as a new high-throughput method to detect the RNA-protein interaction without complicated protein purification steps.

#### 3.1.4. Evaluation of the RNA Binding Affinities and Selectivities of the Designer PPR Proteins

The established RBP-ELISA was used to evaluate the four designer PPR protein genes of the PPR2.0 and SCD scaffolds. RBP-ELISA was performed using the cell lysates, and the equivalent expression of designer PPR proteins was confirmed (Appendix A). The RBP-ELISA assay indicated that all PPR2.0 scaffold PPR proteins displayed higher RNA binding affinities to the target sequence (1.3 to 3.6 times) compared with those of the SCD scaffold (Figure 2a).

RNA binding selectivity was further analyzed using the signal intensity ratio between the target RNA and two non-target RNAs (N1 and N2; Figure 2b). The median ratio was estimated at 200.2 and 53.9 for the PPR2.0 and SCD scaffolds, respectively, indicating significant improvement in the RNA binding selectivity of the PPR2.0 scaffold. These results suggest that the PPR2.0 scaffold in this study displays improved activity in terms of both affinity and selectivity compared to the SCD scaffold and presumably to the available designer PPR proteins also.

#### 3.1.5. Evaluation of the Versatility of the Designer PPR Proteins of the PPR2.0 Scaffold

According to trial experimental results using four designer PPR proteins of the PPR2.0 scaffold (Figure 2), 19 additional PPR proteins were constructed to evaluate the versatility of the PPR2.0 scaffold in this study. In total, 23 PPR proteins were subjected to the RBP-ELISA against all 23 target RNAs for the respective PPR proteins (Figure 3). The results indicated that 21 out of 23 proteins (91%) displayed the highest RNA binding signals on the target RNA. In parallel, the binding affinities of the 12 PPR proteins and the target RNAs were analyzed using EMSA to estimate the apparent *K*_D_ values (Appendix A). The *K*_D_ values were shown to be between 10^−9^ and 10^−7^ M, with 1.95 × 10^−9^ being the lowest score, which was lower than those of the available designer PPR proteins (Table 1).

The *K*_D_ values of the PPR proteins, estimated using EMSA, correlated with the signal values in the RBP-ELISA (R^2^ = 0.74; Appendix A). This suggests that the RBP-ELISA can be used to quantitatively analyze RNA binding affinity in a manner comparable to EMSA. Important issues with RNA manipulation are its applicability to RNA-containing stable structures and unusual sequence context(s), such as high GC content. Therefore, the minimum free energy (ΔG) of the secondary structures of the RNA probe sequences was calculated using RNAfold (http://rna.tbi.univie.ac.at/cgi-bin/RNAWebSuite/RNAfold.cgi; accessed on 20 February 2020). The estimated correlation coefficient between the 70 RBP-ELISA scores and ΔG was low (R^2^ = 0.09; Appendix A), suggesting that the PPR-RNA interaction is less affected by the RNA secondary structure, in contrast to base-pairing-based techniques, such as ASO. To confirm this possibility, the *K*_D_ of the designer PPR protein PPR2.0^t24^, for RNA of low ΔG (−10.85; Appendix A), which forms a stable secondary structure with high GC content (77%), was determined using EMSA (Appendix A). The EMSA showed a high binding affinity at a *K*_D_ of 1.1 × 10^−8^ M, which further strengthened the possibility that designer PPR proteins are applicable for structured RNA and/or RNA in unusual sequence contexts, such as high GC content.

### 3.2. Target Splicing Control Using Artificially Constructed PPR Proteins

#### 3.2.1. Enhancement of Exon Skipping of a Reporter Gene by Designer PPR Proteins in HEK293T Cells

Eukaryotic RNA undergoes various types of RNA processing and modifications. Among these, we focused on splicing to address the application of designer PPR proteins in manipulating the RNA of interest in mammalian cells. It is known that 95% of human genes have two or more spliced forms via complexed mechanisms [4]. Alternative splicing is mediated by the binding of *trans*-splicing factor(s) to *cis*-elements in pre-mRNA. Artificial control of alternative splicing has been achieved by blocking *cis*-elements using various techniques, such as ASO and CasRx [12,33].

In this study, an available reporter system, a bi-chromatic fluorescent reporter plasmid designated RG-6, was used [34]. The reporter gene consisted of three exons. EGFP fluorescence can be visualized when exon 2 is retained, although the fluorescence changes to dsRED by skipping exon 2 (Figure 4a). Exon 2 skipping is facilitated by blocking the polypyrimidine tract. Thus, we selected six target sequences (Figure 4b) in the polypyrimidine tract (sp1–sp4), the intron–exon junction (sp5), and the middle of exon 2 (sp6). The designer PPR genes of the PPR2.0 and SCD scaffolds were constructed by fusing a nuclear localization signal at the N-terminus.

The plasmid of RG-6 and the PPR genes were transiently co-transfected into HEK293T cells, and the splicing pattern change was initially examined by EGFP or dsRED fluorescence using microscopy (Figure 4b). The dsRED fluorescence was significantly increased by transfection of the PPR gene, especially in the designer PPR protein of the PPR2.0 scaffold targeting the sp2 sequence.

However, when the PPR gene was fused to a fluorescent protein gene to analyze cellular localization, unusual distribution or possible aggregation was occasionally observed in the designer PPR proteins of the PPR2.0 scaffold (Appendix A). Therefore, minor modifications were made to the PPR2.0 scaffold, namely, the introduction of basic amino acids at the 16th position (D16K) for the adenine-recognition motifs and the introduction of hydrophilic amino acids at the 7th and 10th positions (L7N, G10Q) of the 1st PPR motif to improve motif-to-motif interaction and solubility, respectively, based on the structural information of the PPR protein [35] (Appendix A). Solubility improvement was verified by gel filtration analysis and imaging analyses using PPR-GFP fusion proteins (Appendix A). In addition, significantly increased RNA binding affinity was observed by analyzing 23 designer PPR proteins using the modified scaffold (Appendix A), although the RNA binding selectivity was equivalent to that of PPR2.0 (Appendix A). Thus, the modified scaffold was designated PPR3.0 and used in subsequent experiments.

When the PPR3.0 scaffold was applied to the RG-6 reporter assay, pronounced improvements in exon 2 skipping were observed by microscopy (Figure 4b). The splicing pattern changes were further analyzed using RT-PCR (Figure 4c). The ratio of splicing variants containing active dsRED coding regions was estimated at 0.48 in normal HEK293T cells. The ratio increased to 0.93 with the addition of the designer PPR gene from 0.48 at the basal level. The medians of the ratios of the dsRED splicing variants were estimated to be 0.57, 0.72, and 0.79 for the SCD, PPR2.0, and PPR3.0 scaffolds, respectively. The results suggest the potential use of designer PPR proteins for splicing control in mammalian cells, and improvements in PPR scaffolds were successfully conducted.

#### 3.2.2. Splicing Control of Endogenous mRNA by Designer PPR Proteins in HEK293T Cells

Toward the end of this study, we applied the PPR technique to endogenous mRNA splicing, choosing *checkpoint kinase 1* (*CHK1*) mRNA as a model molecule. The CHK1 protein is known as a key regulator of the checkpoint signaling in both the unperturbed cell cycle and DNA damage response [36]. Splicing of *CHK1* pre-mRNA is controlled through the cell cycle, and two protein isoforms, CHK1 and CHK1S, are produced in the presence or absence of exon 3, respectively. The highly abundant CHK1S protein inhibits CHK1 function at checkpoint signaling, and overexpression of CHK1S in cancer cells induces cell death [37]. If accumulation of the CHK1S protein can be controlled via exon 3 skipping by the designer PPR protein, cell death induction should be observed.

To control *CHK1* mRNA splicing, the putative enhancer element(s) in exon 3 was computationally searched using a human splicing finder [38], and two target sequences were selected (Figure 5a and Appendix A). The corresponding designer PPR protein genes targeting the 18 nt sequences were constructed using the PPR 3.0 scaffold. RBP-ELISA showed higher binding affinity and specificity for PPR^chk1a^ than for PPR^chk1b^ (Appendix A). Both PPR^chk1a^ and PPR^chk1b^ were transiently transfected into HEK293T cells, and the extent of exon 3 skipping was analyzed using RT-PCR (Appendix A). The designer PPR protein genes of PPR^chk1a^ showed higher activity for exon 3 skipping of the *CHK1* mRNA.

To confirm the change in cell growth and viability by CHK1S induction via splicing control, a stable cell line was established by integration of the PPR^chk1a^ gene under the control of a doxycycline (Dox)-inducible promoter. Two days after Dox induction of the PPR^chk1a^ gene, the splicing extent of *CHK1* mRNA and cell survivability were evaluated. RT-PCR analysis revealed efficient exon 3 skipping, as observed in the transient expression of the PPR gene (Figure 5b). Importantly, cell survivability was significantly decreased after induction of the PPR^chk1a^ gene (Figure 5c). The induction of another designer PPR that targets unrelated RNA has no effect on cell number, indicating that the effect solely depends on the target RNA sequence. In addition, possible off-target effects of PPR^chk1a^ were addressed. A database search revealed the absence of a perfect match or single mismatch sequence in the human genome and 64 genomic loci that contained two mismatch sequences. Although most are non-transcribed regions, two are located within putative non-coding RNA, and the other is located at exon 9 of *murine double minute* 2 (*MDM2*). RT-PCR analysis revealed no splicing abnormality around exon 9 of *MDM2* RNA (Figure 5d), which further strengthened the specificity of the designer PPR proteins in this study. The results clearly demonstrate that the designer PPR protein can be used for RNA control in mammalian cells.

## 4. Discussion

Various RNA manipulation tools have been developed to understand the complexity of RNA regulation, including recently discovered massive RNA splicing and non-coding RNA, which may determine the complexity and identity of mammalian cells, and for practical uses, including therapeutic purposes. Several nucleotide-based technologies, such as RNAi, ASO, and derivatives of CRISPR/Cas, have proven useful in basic and applied biology. Meanwhile, the available protein-based techniques have many restrictions with respect to targeting RNA length and/or sequence context, although high binding affinities and reactivities, various modes of action, and/or low off-target effects are expected from protein-based technologies.

The PPR motif and protein are promising features for designer RNA-binding proteins, with one motif to one nucleotide corresponding to interaction with long RNA sequences and no motif context dependency, whereas selectivity is determined by the particular amino acid position in a programmable manner. In this study, we established a rapid construction and evaluation system and demonstrated the versatility of RNA binding affinity and selectivity. Importantly, this work clearly demonstrated that the designer PPR protein can target long RNA (18 nt) as well as a single specific RNA in the mammalian transcriptome. Furthermore, we demonstrated applications in splicing control in cultured human cells by targeting endogenous *CHK1* mRNA to induce cell death.

To reduce the time and effort required to construct designer RNA-binding proteins, including designer PPR proteins, we built a 96-well-plate-based construction and evaluation system. Using this rapid system, 100 designer PPR proteins were obtained in six days. This system enables the design, construction, and evaluation of designer PPR scaffolds using hundreds of the test PPR molecules. In addition, acquiring biochemical characteristics prior to analysis in cells or individuals would provide significant advantages with respect to understanding the mechanism, which is difficult for nucleotide-based technologies. In principle, the RBP-ELISA system can be applied for the analysis of other RNA- and DNA-binding proteins.

This study also exhibited a high production rate (91%) of designer PPR proteins with expected RNA binding affinities and selectivities. Furthermore, we highlighted the wide range of applications of various RNA species, including RNA with high GC content and highly structured RNA, as shown. When the PPR protein binds to the RNA structure, it is expected that the PPR protein dissociates the structure, since the PPR protein is sequence-specific rather than structure-specific. The plant PPR10 protein has been shown to bind and refold to the secondary structure located in the 5′UTR of *atpH* mRNA [39]. These characteristics would be useful for targeting RNAs that are difficult to target using nucleotide-based technology dependent on RNA-RNA or RNA-DNA base pairing. In addition, the PPR protein can target any RNA sequence apart from the limitation of the PAM sequence of CAS13d/CasRx [11,12].

Alternative splicing is a hallmark of mammalian transcriptomes. The mechanism remains, but it is widely held that binding of the *trans*-splicing factors to *cis*-elements on the pre-mRNA can mediate exon inclusion, exon exclusion, or the creation of alternative splice sites [40]. In the case of the bi-chromatic fluorescent reporter plasmid (RG-6) used in this study, it is known that skipping the cTNT exon corresponds to exon 2 of the RG-6 reporter, which is induced by MBNL protein binding to the polypyrimidine tract and inhibiting binding of the splicing factor (U2) [41]. Indeed, the designer PPR protein targeting the sp1 region was highly efficient in facilitating the exon 3 skipping from 48% basal to 93% (Figure 4). In contrast, a previous study using the same RG6 reporter system and dCasRx showed the highest exon skipping by targeting the splice acceptor, which is equivalent to the region sp5, from 8% basal to 65% for dCasRx alone and 75% with hnRNPa1 fusion [12]. This suggests that the designer PPR protein has an exon skipping efficiency comparable to that of the dCasRx system. In addition, the designer PPR protein also facilitated exon skipping, and improvement of the PPR scaffold from PPR2.0 to 3.0 strengthened exon skipping in various target regions, suggesting less *cis*-element dependency. Further analyses will be required to evaluate the efficiency and off-target effects of designer PPR proteins and the nucleotide-based method in parallel, as well as to further understand alternative splicing.

In this study, the sole PPR motif repetition was applied to the splice control, as in ASO. Various modes of action using designer PPR proteins would be expected by fusing the effector protein domain, such as translational enhancement, which we have previously shown using a fusion protein of natural PPR protein of CRR4 and a partial eIF4G domain in HEK293T cells [42]. Similar strategies have been demonstrated for the Pumilio protein in RNA degradation, RNA visualization, and translational activation by fusing RNase, GFP, and poly(A)-binding protein (PABP), respectively [19,20,21]. These applications can be applied to the designer PPR proteins. It has been demonstrated that several classes of natural PPR proteins provide a C-to-U RNA editing enzymatic domain at the C-terminus, and the function was verified by reconstitution experiments in *E. coli* [43,44]. Most recently, we identified U-to-C RNA editing enzymes and demonstrated their reconstitution in bacteria and cultured human cells [45]. Fusing the RNA editing domain with the designer PPR protein and its use as a single-RNA-base-substitution tool is a promising application.

In summary, designer PPR proteins would provide a simple and scalable option to study and manipulate RNA, in addition to the RNA-targeting CasRx system. Although the protein-based tools, including designer PPR proteins, still require complicated procedures for their use compared to the RNA-targeting CRISPR/Cas system, we would anticipate attractive advantages for protein-based tools, as seen in various catalytic reactions in which nucleotides have been replaced with proteins during biological evolution. Further analysis is required to address the advantages and disadvantages of protein-based and nucleotide-based tools and to determine their appropriate usages depending on applications. We hope that the new protein-based RNA engineering tool presents a new frontier in biology, as do genome editing tools, such as zinc-finger, TALE, and CRISPR/Cas9.

## 5. Patents

Kyushu University and EditForce, Inc., have filed patents regarding an efficient method for the construction of the PPR gene (PCT/JP2020/021472) and improvement of the PPR scaffold to increase its solubility (PCT/JP2020/021473). Y.Y., M.T., T.I. and T.N. are the inventors listed in the patents.

## Figures and Tables

**Figure 1 cells-11-03529-f001:**
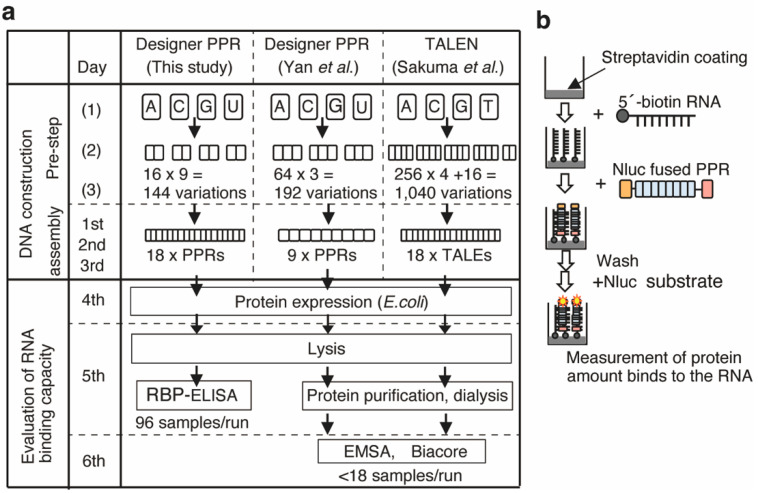
Fast assembly and evaluation system for the designer PPR protein. (**a**) Scheme for seamless repeat assembly of the PPR ([32] and this study) and TALEN [30] genes and evaluation of the RNA binding affinity and specificity using RBP-ELISA for the recombinant PPR protein. (**b**) Experimental flow of the RNA-binding protein ELISA (RBP-ELISA).

**Figure 2 cells-11-03529-f002:**
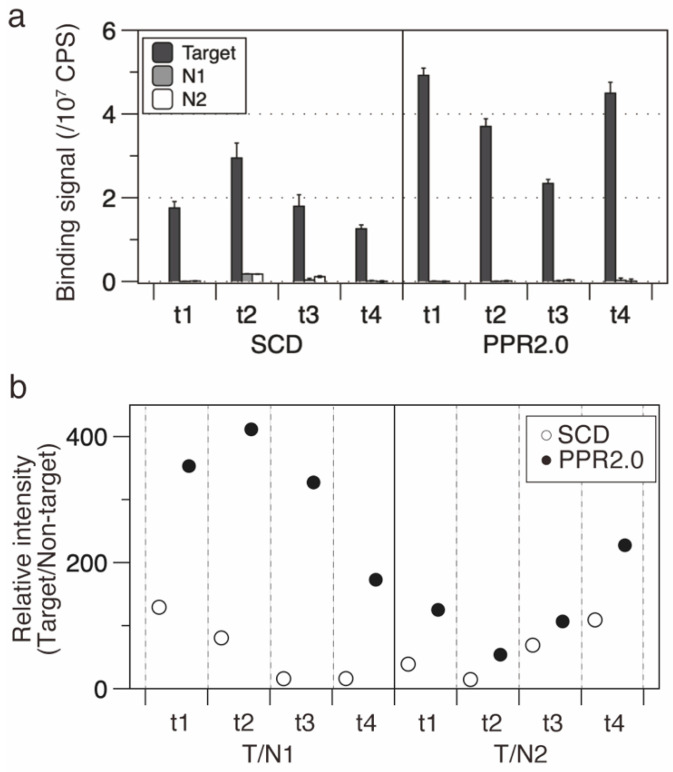
Designer PPR proteins of the PPR2.0 scaffold have higher binding activities and selectivities than those of the SCD scaffold. (**a**) RBP-ELISA for SCD and PPR2.0 scaffold proteins. Four PPR proteins were subjected to analysis using their respective target sequences (t1, t2, t3, and t4) or two non-target sequences (N1 and N2). (**b**) RNA binding selectivity. The selectivities of the SCD and PPR2.0 scaffolds were examined and plotted by estimating relative intensities, which are the signal intensities of target RNA against those of non-target RNA.

**Figure 3 cells-11-03529-f003:**
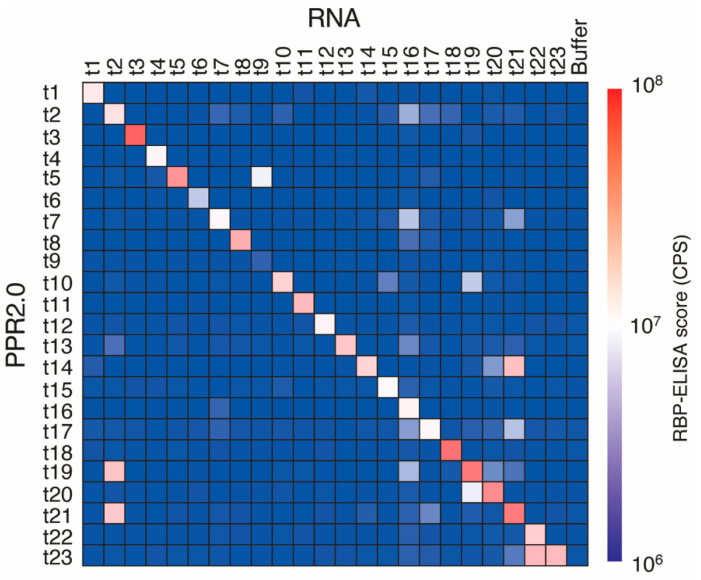
Evaluation of the versatility of the designer PPR proteins of the PPR2.0 scaffold. Heatmap of the binding signals of the designer PPR proteins. RBP-ELISA was used to perform all combinations (288 reactions) using 23 designer PPR proteins (t1 to t23) and their target RNAs, with no RNA (buffer) as a background signal.

**Figure 4 cells-11-03529-f004:**
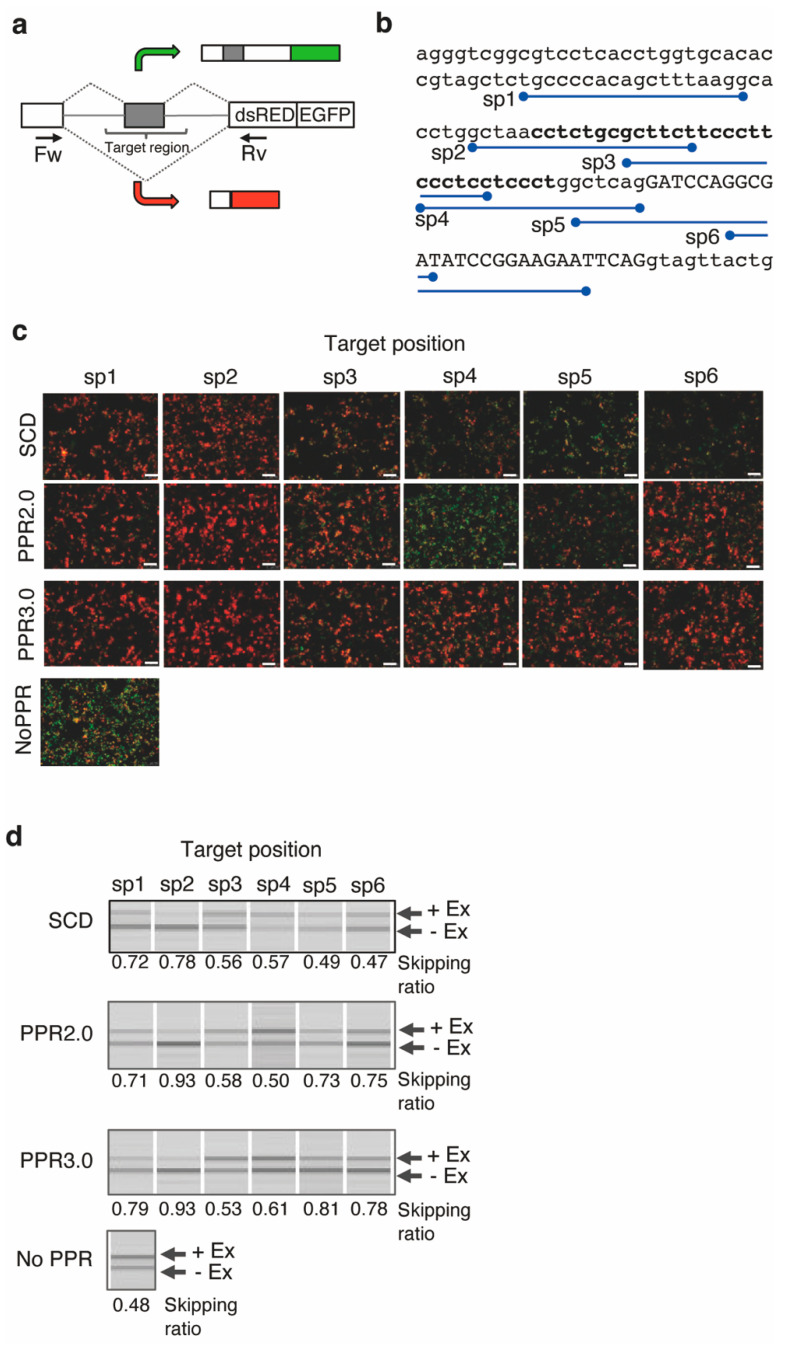
A reporter assay for splicing control by designer PPR proteins. (**a**) Diagram of the reporter system RG-6. The bi-chromatic alternative splicing reporter transcribes RNA containing the EGFP or dsRED coding frame in the presence or absence of exon 2. (**b**) Sequence of the target region. The positions of the target sequences (sp1–6) for the designer PPR proteins are depicted. The exons and introns are shown by uppercase and lowercase letters, respectively. The polypyrimidine tract is highlighted using bold letters. (**c**) Cell image analysis for the splicing control. Merged fluorescent images of GFP and dsRED show the designer PPR proteins of the SCD, PPR2.0, or PPR3.0 scaffolds transfected into HEK293T cells with RG-6 reporter plasmids. The control experiment was performed without transfection of the PPR gene. The white bar indicates 10 µm. (**d**) RT-PCR analysis of the splicing variants. The extent of alternative splicing (presence or absence of exon 3; +Ex, −Ex) was analyzed using RT-PCR with forward and reverse primer sets, as shown in (**a**). The skipping ratio was estimated by calculating the signal intensity of [−Ex]/([+Ex] + [−Ex]).

**Figure 5 cells-11-03529-f005:**
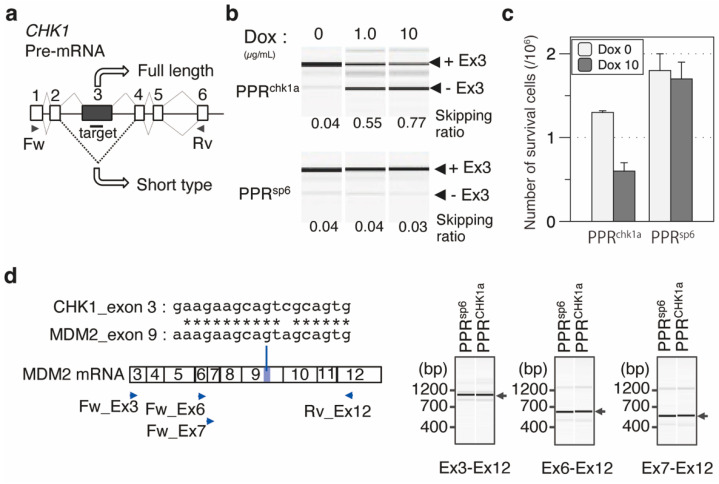
Designer PPR proteins induced exon skipping of *CHK1* mRNA and the retardation of cell growth. (**a**) Exon–intron structure of *CHK1* pre-mRNA. The target position was designed in exon 3 of the *CHK1* gene. (**b**) RT-PCR analysis of exon skipping. The amount of the splicing variant with or without exon 3 (+Ex3 or −Ex3) was analyzed using Dox-induced expression of PPR^chk1a^. PPR^sp6^ was used as a negative control. The primer position used for RT-PCR is shown in (**a**). The skipping ratio was estimated by calculating the signal intensity of [−Ex3]/([+Ex3] + [−Ex3]). (**c**) Analysis of cell number with or without the induction of the designer PPR genes (PPR^chk1a^ or PPR^sp6^). Cell number was analyzed using trypan blue staining with the cells two days after dox induction (N = 3). (**d**) Off-target RNA analysis. Two mismatched chk1a sequences located in MDM2 exon 9. The exon was confirmed using RT-PCR with Ex3–Ex12, Ex6–Ex12, and Ex7–Ex12 primer sets. Primer positions are indicated by blue triangles.

**Table 1 cells-11-03529-t001:** Amino acid sequences of PPR scaffolds reported in previous published papers and their binding affinities to target RNA sequences.

Name	Number of PPR Motifs	Binding Affinity ^1^ (KD)	Sequence of PPR Scaffold	Reference
PPR2.0_A	18	This study	VVTYTTLIDGLCKAGDVDEALELFKEMRSKGVKPN	This study
PPR2.0_C	18	This study	VVTYNTLIDGLCKSGKIEEALKLFKEMEEKGITPS	This study
PPR2.0_G	18	This study	VVTYTTLIDGLCKAGKVDEALELFDEMKERGIKPD	This study
PPR2.0_U	18	This study	VVTYNTLIDGLCKAGRLDEAEELLEEMEEKGIKPD	This study
dPPR SCD	10	90 nM	VVTYXTLIDGLCKAGKLDEALKLFEEMVEKGIKPX	[26,27,28]
	11	7.5 nM	VVTYXTLISGLGKAGRLEEALELFEEMKEKGIVPX	
	14	16 nM	VVTYXTLIDGLAKAGRLEEALQLFQEMKEKGVKPX	
cPPR	8	370 nM	VVTYTTLIDAFCRKGRLDEALSLFSEMKSKGIKPN	[25]
*synth*PPR	4	No data	VVTYNTLISGFCKAGRLEEAMSLFSEMKSKGLVPS	[29]
MCD_A	14	18 nM	VVTYTILIDALCKAGRLEEALSLFSEMKEIGIKPD	[28]
MCD_C			VVTYNILIKGLCKAGKLEEALSLLSEMVEKGIQPD	
MCD_G			VVTYNTLIDGLCKSGKIEEALKLFKEMEEKGITPS	
MCD_U			VVTYTTLIDGLCKAGKVDEALELFDEMKERGIKPD	

^1^ Binding affinity is estimated by EMSA.

## Data Availability

Not applicable.

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
