# Peer review of "Construction of a Versatile, Programmable RNA-Binding Protein Using Designer PPR Proteins and Its Application for Splicing Control in Mammalian Cells"

_cells, 2022, doi:10.3390/cells11223529_

Round 1

Reviewer 1 Report

The idea of RNA editing as a technique to manipulate molecular mechanisms of RNA functioning, like modulation of splicing and others. Authors propose to use PPR proteins, which may be programmed to the tight interaction with specific RNA sequence. 

Investigation contains a lot of technical information and demonstrates huge work performed by authors to generate a set of genetic constructs for expression of pool of PPR proteins designed for binding with specific RNA sequence. 

The view of the data presenting RT-qPCR results look artificial and it will be better to demonstrate real gels but not edited pictures. 

Despite the douptful perspective of the real usage of the proposed RNA-editors because of very complicated procedure of creation the idea to generate some tool for the manipulation with exome not at the level of genome but at the RNA level looks very attractive.

Author Response

We appreciate your comments. The responses is as follows.

Comment 1: The view of the data presenting RT-qPCR results look artificial and it will be better to demonstrate real gels but not edited pictures.

Response: As you pointed out, we have used an automatic capillary electrophoresis system that provides artificial gel images. However, the capillary system would be better than real gel analysis for quantitative analysis of the splice variant. Therefore, we have decided not to change this figure.

Comment 2: Despite the doubtful perspective of the real usage of the proposed RNA-editors because of very complicated procedure of creation the idea to generate some tool for the manipulation with exome not at the level of genome but at the RNA level looks very attractive.

Response: Thank you for your suggestion. We have revised the last part of the discussion to highlight the advantages and disadvantages of protein-based tools versus nucleotide-based tools, and stressed upon the need of further experiments in the near future (P15, L578-586).

Reviewer 2 Report

In this manuscript, Yagi et al.designed a set of PPR motif based proteins to affect cellular splicing.In general,this is an interesting and timely investigation, especially for the development of protein basd RNA manipulation technology. However, the following questions need to be addressed.

(1)The PPR was orignially derived from plant protein, are they conserved through the species? does the author use the plant or human homolog in their experiment? why?

(2)It's generally thought the binding affinity between RBP and RNA sequences is modest and less specificity,when compared with base pairing between nucleotide or protein-protein interaction. Do the authors compared their results with an artificial splicing systems with nucleotide based method?

(3) To increase the specificity, you've produced desinger PPR protein with multiple motifs. It's seemed that there are difficulties in DNA cloning and protein production steps( as described in line 264-265). Further,there are also complicated RNA binding affinity selection process.Finally, the engineered protein may have limited delivery or packaging efficiency.So it is still too early to say PPR based method is better than CRISPR/Cas-based systems.

Based on above issues, I'd like to see additional experiments or an in-depth discussion to clarify them. 

Author Response

We appreciate your comments. The responses are as follows.

Comment 1: The PPR was originally derived from plant protein, are they conserved through the species? does the author use the plant or human homolog in their experiment? why?

Response: In this study, we used 322 PPR motifs in Arabidopsis to generate a semi-consensus PPR scaffold. The corresponding sentence has been revised (P7, L281&282).
 Dicots and monocots contain 500 PPR genes, and 80% of PPR genes are homologs. Humans have fewer than 10 PPR genes. This was too small to generate a semi-consensus PPR scaffold. We did not incorporate the above two pieces of information because they may not be apart from the focus of this manuscript.

Comment 2: It's generally thought the binding affinity between RBP and RNA sequences is modest and less specificity, when compared with base pairing between nucleotide or protein-protein interaction. Do the authors compared their results with an artificial splicing systems with nucleotide based method?

Response: We have only compared the results with previous literature on dCasRx using the RG6 reporter (Figure 4; P14, L554 to 556). The sentence was revised to highlight the necessity of evaluating the advantages and disadvantages of the designer PPR protein against the nucleotide-based method (P14, L556 to P15, L558; P15, L561 to 562).

Comment 3: To increase the specificity, you've produced designer PPR protein with multiple motifs. It's seemed that there are difficulties in DNA cloning and protein production steps( as described in line 264-265). Further, there are also complicated RNA binding affinity selection process. Finally, the engineered protein may have limited delivery or packaging efficiency. So it is still too early to say PPR based method is better than CRISPR/Cas-based systems.

Response: We agree with this comment. We have revised the end of the discussion to highlight the advantages and disadvantages of protein-based tools and nucleotide-based tools, and stressed upon the need of further experiments in the near future (P15, L578 to 584).

In addition, the manuscript was subjected to a professional English editing service to respond to your comment “Moderate English changes were required”.

Round 2

Reviewer 2 Report

It's OK for publication now.